# "They are hard to navigate" exploring healthcare providers experiences managing Self-Injurious Behaviors among children with Autism Spectrum Disorder in Uganda

Hillary Mugabo Mukula[1,2], Harriet Opondo[1], Morris Ndeezi[1]*, Anita Kateregga[1], Khamisi Musanje[3], Samuel Ouma[1]

**1** Department of Mental Health and Community Psychology, Makerere University, Kampala, Uganda, **2** Department of Counseling and Higher Education, Ohio University, Athens Ohio, United States of America, **3** Department of Educational, Social and Organizational Psychology, Makerere University, Kampala, Uganda

\* morrisndeezi@gmail.com

## Abstract

Self-injurious behaviors (SIBs), although not a symptom of Autism Spectrum Disorder (ASD), are highly prevalent among children with ASD and present significant management challenges for healthcare providers, particularly in low-resource settings. This qualitative study explored the experiences of healthcare providers managing self-injurious behaviors in children aged 5–12 years diagnosed with Autism Spectrum Disorder. Ten healthcare providers were recruited using maximum variation purposive sampling. Data were analyzed thematically using Braun and Clarke's framework. Two main themes emerged from the data (1) Experiences managing SIBs and (2) challenges managing SIBs. Findings suggest challenges in managing self-injurious behaviors, which are exacerbated by a lack of specialized healthcare providers and the limited use of evidence-based interventions. Targeted support, specialized training, and supervision are recommended for healthcare providers.

## Introduction

Self-Injurious Behaviors (SIBs), though not a symptom of Autism Spectrum Disorder (ASD), are highly prevalent among children with ASD [1]. Currently, the prevalence of SIBs among children with ASD is estimated at 42% [2]. SIBs, also known as *self-harm, self-mutilation, and non-suicidal self-injury (NSSI)* in the literature, refer to behaviors where individuals harm themselves, causing physical injury or tissue damage [1,3,4]. Common forms of SIBs in children diagnosed with ASD include head banging, self-cutting, self-choking, self-biting, hair pulling, skin picking, and many others [5,6]. A recent meta-analysis revealed that the estimated pooled prevalence of SIBs, such as hand hitting (23%), skin picking (approximately 20–25%), and hitting oneself against objects (15%), are the most common [7].

**Data availability statement:** key data are presented within the manuscript, and any additional supporting data have been uploaded as supplementary information accompanying the submission.

**Funding:** The author(s) received no specific funding for this work.

**Competing interests:** The authors have declared that no competing interests exist.

Managing SIBs in children with ASD is especially difficult in low-resource settings because of limited healthcare infrastructure and competing health priorities [8]. Traditional approaches such as Applied Behavior Analysis [ABA], Cognitive Behavioral Therapy [CBT], Dialectical Behavior Therapy [DBT], and structured behavior plans, which are common in high-income countries, are often unavailable or ineffective in low-resource settings. Additionally, these strategies require highly trained professionals, long-term therapy, and are often unaffordable for most families, with the cost of therapy often covered by parents and caregivers [9,10]. The shortage of trained specialists, combined with widespread poverty, contributes to limited access to adequate ASD and SIBs care in low-resource settings, including Uganda.

In Uganda and similar low-income settings, healthcare providers (HCPs), such as pediatricians, psychologists, and speech and occupational therapists, are the first line of referral for children with SIBs [11]. However, these providers often lack specialized training in ASD and the management of SIBs, leaving a critical gap in service delivery. The limited number of professionals with relevant expertise creates barriers in care, and those who are available face extensive caseloads, exhaustion, and burnout [12]. Additionally, the social and cultural beliefs about disabilities in Uganda make the treatment and management of SIBs even more complicated [13].

Existing research from high-income countries has examined healthcare providers' experiences supporting children with ASD; some of the challenges highlighted included limited autism-specific training, communication barriers, and systemic constraints within healthcare settings [14,15]. In African contexts, there remains limited literature, but emerging studies from Uganda, Nigeria, and South Africa have documented HCPs knowledge gaps, attitudes, and challenges related to ASD assessment and support [16–18]. However, none of this literature focuses on the experiences of HCPs managing SIBs among children with ASD.

The absence of specialized training for managing children with SIBs in Uganda implies that HCPs must often be creative and rely on their experiences to support their clients. Given these realities, it is critical to explore the experiences of HCPs managing SIBs among children with ASD in Uganda. These professionals are central to diagnosis, intervention, and family support, yet their voices are often underrepresented in research. Understanding their experiences can inform the development of targeted training programs, supportive policies, and culturally appropriate intervention strategies.

## Materials and methods

### Design

This study used a qualitative phenomenological design to explore healthcare providers' experiences managing SIBs among children diagnosed with ASD. Data was collected using in-person semi-structured interviews. The study was conducted in the Central Region of Uganda, particularly in Kampala. The central region is highly diverse and presents an appropriate setting for this study, as studies have shown that most specialized healthcare providers in Uganda are located within the central

region [19,20]. Furthermore, the central region has been found to have the highest population of children diagnosed with ASD [21].

## Participants, sampling, and eligibility

We used maximum variation purposive sampling to recruit participants from diverse settings within the central region of Uganda. The central region is highly diverse and is home to Uganda's capital city, Kampala. Recruitment was conducted from 24th June 2024 to 20th August 2024 through the dissemination of study flyers in HCPs' professional associations, including the Uganda Occupational Therapy Association (UOAT) and the Allied Health Professionals Council of Uganda (AHPCU), and through participant referrals. To disseminate the study flyers within professional associations, permission was sought from the secretary general of each association, who later forwarded the flyers to members. Thirteen health-care providers expressed interest in participating in the study, including pediatricians [2], a psychiatrist, clinical psychologists [3], Occupational Therapists [3], and speech and language therapists [4]. Ten participants met the inclusion criteria for this study. To ensure variation, the sample was varied by professional background, years of experience, training, and work settings.

To be eligible for participation in the study, HCPs must have [1] had experiences supporting children diagnosed with ASD and presenting with SIBs, [2] possessed a minimum educational qualification equivalent to a diploma, [3] the ability to read and write in English, and [4] be capable of providing consent. HCPs who did not meet the above criteria were excluded.

## Data collection

A semi-structured interview guide was used to allow the emergence and free flow of participant experiences. The interview guide was developed by the principal investigator (HMM) in consultation with the co-investigators (HO and SO). The questions in the interview guide were informed by previous research and the clinical experiences of the researchers. Sample questions from the interview guide are included in **Table 1** below.

**Table 1. Semi-structured interview guide.**

| Topic | Sample questions |
| --- | --- |
| Grand tour question | Tell me about your experiences working with children diagnosed with Autism spectrum disorder? |
| | How long have you been working with children with ASD? |
| Experiences managing SIB | What are your experiences managing SIBS among children with ASD? |
| | Do you have any specialized training in managing self-injurious behaviors (SIBs)? |
| Common forms of SIBs | What are some of the common forms of SIBs that you have managed? |
| | Can you describe how these forms of self-injurious behavior (SIBs) manifest? |
| Management strategies | What are some of the strategies you used to manage the SIBs you encountered? |
| | What are some of the challenges you faced while implementing these strategies |
| | What were your experiences working with parents and caregivers? |
| Management recommendations | What can be done to improve the management of SIBs in Uganda? |

We conducted interviews between June and August 2024. The principal investigator (HMM) verified participant eligibility, explained the study procedures, and obtained written informed consent from eligible participants. Four participants received the consent form via email before the interview, while six received it on the day of the interview. Data were collected through face-to-face, in-depth interviews using a semi-structured interview guide. Each interview lasted approximately one hour and was conducted in English. Four interviews were conducted at the participants' workplaces. Prior to conducting any interviews at the participants' workplaces, the researchers ensured that the participants had permission from their heads of department and felt safe in their office spaces. Six interviews were conducted at the Principal Investigator's office. The interviews were conducted separately by HMM and the Co-Investigator (HO). The researchers had no prior relationship with the participants, and all interviews were conducted in secluded rooms with only the interviewer and the participant present. Interviews were conducted until data saturation was reached; at that point, no new insights relevant to the research question emerged from additional interviews.

### Ethical consideration

This study was reviewed and approved by the Makerere University School of Health Sciences Research and Ethics Committee (MAKSHSREC-2024–735). To ensure voluntary participation, participants were provided with key information about the study through the consent form. To participate, participants had to provide their consent after reading the consent form, ensuring understanding of the objectives, procedures, and their rights. All participants received Uganda Shillings 20,000 as compensation for their participation in the study.

### Data management and analysis

All interviews were recorded and transcribed verbatim by the principal investigator (HMM), who listened to each recording to ensure accuracy. All transcripts were shared with the participants for member checking, and the analysis process only began after the participants confirmed that their perspectives and experiences were accurately captured. Data coding and initial analysis were conducted by HMM, HO, and MN, both manually and using NVivo, a qualitative data analysis software. The thematic analysis steps outlined by [22] were followed. First, HMM, OP, and NM randomly picked five transcripts for initial analysis. We then read and re-read the transcripts to familiarize ourselves with the data. After immersing ourselves in the data, we separately identified common patterns and quotes, and then developed codebooks. Next, we met to discuss our findings and agreed on what to include, creating a codebook (S1 Code) upon which the remaining five transcripts would be applied. Throughout the process, thematic statements and quotes that reflected HCPs experiences were identified and isolated.

To ensure trustworthiness of the findings, the authors employed the four criteria outlined by [23]. First, the credibility of our findings was ensured through member checking, whereby participants reviewed and confirmed the accuracy of their transcripts, and through an independent audit conducted by co-investigator SO, a lecturer with expertise in qualitative data analysis, who reviewed the codebook and transcripts to verify the accuracy of our analysis. Secondly, the dependability of our findings was ensured through the step-by-step documentation of the analysis process. During the analysis and writing phase, the authors held regular meetings and maintained an audit trail that enabled all decisions to be traced and reviewed. Transferability was enhanced by providing a thick description of the study context, characteristics of the participants and the themes identified to enable ease of applicability, among future researchers. Finally, confirmability was achieved by the researchers being careful not to allow their prior experiences interfere with the interviewing and data analysis process. The research team met regularly during the writing phase to ensure fidelity to the steps of thematic analysis outlined by [22].

## Results

### Participants

A total of ten HCPs (three males and seven females) participated in this study. Participant demographics are summarized in **Table 2**. The majority of the participants were recruited from school settings, accounting for 50%. Additionally,

**Table 2. Participants demographics.**

| Participants | Sex | Work setting | Educational level | Years of experience | Number of children with SIBs seen over time | Specialized training in ASD treatment and management |
|---|---|---|---|---|---|---|
| Speech and Language Therapist | Female | School | Bachelors Degree | 3 years | 8 | No |
| Occupational Therapist | Male | Rehabilitation center | Diploma | 8 years | 2 | No |
| Occupational Therapist | Female | School | Diploma | 7 years | 6 | Yes |
| Occupational Therapist | Male | School | Masters | 9 years | 6 | No |
| Clinical Psychologist | Female | School | Masters | 9 years | 3 | Yes |
| Clinical Psychologist | Female | Hospital | Masters | 8 years | 3 | No |
| Clinical psychologist | Male | Hospital | Masters | 12 years | 5 | No |
| Pediatrician | Female | Hospital | Doctorate | 20 years | >30 | No |
| Speech and Language Therapist | Female | School | Bachelor's degree | 6 years | >10 | No |
| Psychiatrist | Female | Private Practice | Master's degree | 4 years | 8 | No |

Occupational Therapists and Clinical Psychologists comprised the majority of the study participants, with 30% each. The majority of the participants did not have specialized training in managing ASD and SIBs specifically.

As part of the demographic survey, participants in this study were asked to identify the common self-injurious behaviors they have encountered in their professional practice. **Table 3** indicates the types of SIBs reported by the participants.

Our findings illuminate the meanings and experiences of HCPs managing SIBs among children diagnosed with ASD. Overall, the management of ASD and SIBs is a complex process for many HCPs in Uganda. Two themes emerged from the data: [1] experiences managing SIBs and [2] Challenges managing SIBs. A summary of the themes, subthemes, and codes is provided in Table 4 below.

**Personal interpretation of SIBs and their triggers.** This sub-theme captures how HCPs perceived the manifestation of SIBs and their associated triggers among children with ASD. Codes within this sub-theme included communication, self-stimulation, repetitive behavior, emotional regulation, and task escape.

HCPs commonly described SIBs as a means of communication, particularly among non-speaking children or those with limited expressive language. Providers noted that children often engage in SIBs to express needs, frustration, or discomfort when they are unable to communicate verbally:

*"I think the major cause is low or a lack of communication skills. Because ideally, where another child would say, 'I do not want to go,' or 'I do not want to eat this,' this child does not have a means of communicating their needs."*- **Psychologist 2**

*"Most of these self-injurious behaviors I have noticed are among non-verbal children…they cannot tell you, 'I am tired.' That is a way of communicating for them."* – **Occupational Therapist 1**

*"…they commonly occur when the children are in distress…or want to communicate something but are not able to."* **Psychiatrist**

Providers highlighted self-stimulating as being central to the development of SIBs. Notably, they identified boredom as contributing to the persistent SIBs because most of these children have nothing to do, as typified in the quotes below

**Table 3. Common self-injurious behaviors managed.**

| Healthcare Provider | Self-injurious behavior (s) managed |
|---|---|
| Speech and Language Therapist- 1 | Head banging, self-hitting, hair picking/pulling, head slapping, and hair-eating |
| Occupational Therapist-1 | Head banging, self-hitting, self-biting, hair picking, eye gouging, and head slapping |
| Occupational Therapist-2 | Head banging, self-biting, self-scratching, and head-slapping |
| Occupational Therapist-3 | Head banging, self-hitting, self-biting, skin picking, hair-pulling, self-scratching, eye gouging, and head slapping |
| Clinical Psychologist-1 | Head banging, self-hitting, and head-slapping |
| Clinical Psychologist-2 | Head banging, self-biting, skin picking, hair-pulling, self-scratching, and head slapping |
| Clinical Psychologist-3 | Head-banging, self-hitting, skin-picking, and head-slapping |
| Pediatrician | Head banging, self-hitting/pinching, self-biting, skin picking, self-scratching, and head-slapping |
| Speech and Language Therapist- 2 | Head banging, self-hitting, self-biting, skin picking, hair-pulling, self-scratching, head-slapping, and hair-eating. |
| Psychiatrist | Head banging, self-hitting, self-biting, and skin picking |

"…*they were not being engaged, so as a way to occupy themselves, they keep pinching themselves*." **Occupational Therapist 1**

"*Sometimes I think it is probably because of boredom or trying to self-stimulate when they have nothing to do.*" **Psychiatrist**

Some providers further noted that they perceive SIBs among children with ASD to be a result of the repetitive and stereotypic behaviors that are common among children with ASD. Providers narrated that the repetitive nature of SIBs fits into the repetitive behavioral symptoms of ASD, as illustrated in the quotes below,

"*The repetitive nature…when they start this self-injurious behavior, somehow, they fit into that repeated kind of routine, which is purposeless*." **Psychologist 1**

""*Not that they want to hand-bang, but it is the repetitive behavior. It starts with banging on softer surfaces and then really banging on hard surfaces.*" **Pediatrician**

Task escape was also identified as another cause for SIBs in children with ASD. Providers described escape as mainly occurring when children are engaged in activities they find difficult, as shown in the quotes below

"…*if an activity is hard…he would start hitting himself…they realize that this is the way for them to escape.*" **Speech and Language Therapist 1**

"…*they occurred most times when there are changes in routine or when the child wants to run away from a particular environment. They could have those temper outbursts, and they could trigger that to happen*" **Occupational Therapist 3**

**Table 4. Summary of themes, sub-themes and codes.**

| Themes | Sub-themes | Codes |
|---|---|---|
| Experiences Managing SIBs | Personal interpretation of SIBs and their triggers | Means of communication |
| | | Self-Stimulation |
| | | Repetitive Behavior |
| | | Task Escape |
| | Techniques and strategies used to manage SIBs | Pharmacologic interventions |
| | | Behavioral interventions |
| | | Use of protective aids |
| | | Environmental modification |
| Challenges managing SIBs | Provider related challenges | Limited knowledge and skills |
| | | Inadequate resources |
| | | Provider frustrations |
| | | Difficulties working with parents |
| | Parent related challenges | Financial constraints |
| | | Stigma |

Perspectives reflected in this sub-theme align with behavioral and ecological frameworks, suggesting that SIBs serve communicative, self-stimulatory, and escape functions, particularly in contexts of limited stimulation. This highlights the role of both individual deficits and environmental factors in the persistence of SIBs.

**Techniques and strategies used to manage SIBs.** Under this sub-theme, we explore the different techniques and strategies providers use to manage SIBs in children with ASD. Providers described various techniques utilized in the management SIBs, including the use of medicines and other behavioral approaches. They also highlighted the importance of protective aids and environmental modifications as key components of their management strategies.

While reflecting on techniques and strategies they utilize when managing SIBs among children with ASD, providers identified using medicines as one of the main approaches. They noted using medicines to help calm the child or to address co-occurring conditions like Attention Deficit Hyperactivity Disorder (ADHD), as hereto stated.

*"Usually, the time when I have to go the route of medicine is generally for those who have Co-morbidities. Let's say they have self-injurious behavior, but maybe they also have ADHD. Usually, we use antipsychotics, Risperidone, on a low dose, and see how that works."*- **Psychiatrist**

*"What I have seen in the children's clinics is some children with those challenging behaviors they are given medication to first calm them down a bit. Most of the time, I see they are given a certain medicine, Risperidone."*- **Occupational therapist 1**

Providers described other techniques they use to manage SIBs, these were typically behavioral strategies and centered on improving the child's communication, regulating emotional distress, reinforcing positive behaviors, and reducing environmental stressors. Providers described interventions such as the use of diversional and reinforcement techniques, play and art therapy techniques, skills training, and, in some cases, the use of physical restraint when children posed an immediate danger to themselves or others around them, as reflected in these excerpts.

*"Restraining is one of the things that we would do, you sit behind the child, and you hold them and then you sing, and distract them."* - **Psychologist 2**

*"What I do most of the time is restrain. Restraining the hands, trying to calm the child down, and then trying to draw them back to the activity, because time-out doesn't really work."* **Speech and Language Therapist 1**

Furthermore, providers described the use of social skills training as one of the strategies they utilize to manage self-injurious behaviors. They described social skills training aspects, including turn-taking and social stories, as reflected in the quotes below,

*"We teach a lot of social skills, especially taking turns, and also help them to know how to wait for something."* – **Occupational therapist 2**

*"One of the key things when I am working with children with autism is social skills training. So, we do lots of social stories, depending on the age. We do experiential social stories."*-**Psychologist 3**

Providers further indicated of redirection techniques as one of the creative strategies they employed to divert children from engaging in SIBs. Diversional techniques were mainly used to redirect the child to a less harmful behavior, as indicated in the excerpts below,

*"I also use diversional therapy [re-directions activities], like you engage the child in another activity, you change their focus. You redirect them to another activity, such as playing with sand. It serves as a good sensation for them and a meaningful activity."*- **Occupational Therapist 1**

*"For the pinching, what I mostly did that worked was distraction. He used to love horses so, much…I would quickly redirect him to a horse. There was a toy horse in the corner, which he would sit on and pretend that he was riding."*- **Psychologist 2**

Providers also shared the use of protective gear as being essential for managing SIBs, especially in extreme cases. Working in low-income settings, providers mentioned adapting different materials and using them as protective gear. These included socks, which they used as gloves.

*"I encouraged her to make sure they wear gloves. But she could not afford them, so she used socks. Just to prevent this child from scratching themselves."* –**Psychiatrist**

*"…he would bite all the nails; they were completely off. Now he started eating the skin of the palm, and it could bleed dry, to the extent that the palms became so hard and it was like the palms of a builder. So what was the first solution, gloves. He puts on gloves every time he eats food, gloves after showering in the evening. That is what we would do."* – **Occupational Therapist 2**

In addition, providers reported utilization of splints as a form of protective gear specifically to manage various forms of SIBs including head banging, self-pinching, and punching, as illustrated below,

*"Sometimes, if a child is injuring themselves, they are injuring their hands or they are injuring their body using their hands, we started splitting them, especially that hand, because when you split that hand and it is not flexing, then it cannot make the fist."*- **Psychologist 1**

*"For the one who used to pinch, I put masking tape on his arms so that when he pinches, because he used to really pinch hard, and you would see flesh, like a fresh wound. So, we cut off his nails, and we also put masking tape."* - **Psychologist 2**

Additionally, providers indicated environmental modification as strategy to manage SIBs. Providers identified various items within the child's immediate environment that the child can potentially contribute to self-harm. These items were either modified or removed from the child's environment to ensure safety.. Environmental modification was utilized as a precautionary measure, as highlighted in participant's comments below,

*"We cushion the room, cushion the floors, put those soft cushions. If the child wants to hit themselves, there is a soft thing where they can do that, not injurious."* – **Occupational Therapist 1**

*"I stopped working with them from places that had chairs and tables, and would go to an empty room that had nothing but floor padding. So that even if they threw themselves down, even if they banged their heads on the floor, I knew that it was safe.* **– Psychologist 2**

**Challenges in managing SIBs.**  Under this theme, HCPs described challenges encountered when managing SIBs in children with ASD. These challenges range from providers' frustration due to the lack of progress despite trying different strategies to frustration caused by parental or caregiver inconsistencies in implementing providers' recommendations. We classified these challenges into two categories: (i) provider-related challenges and (ii) parent-related challenges.

Findings under the provider-related challenges sub-theme included limited knowledge and skills, inadequate resources, frustrations, and difficulties working with parents. Providers described skills and knowledge gaps to adequately support children presenting with SIBs. Some reported that although they learned about ASD in their training, they felt ill-prepared to effectively manage SIBS within clinical settings.

*"We are not really trained in detail… we met these behaviors in the field."* – **Speech Therapist**.

*"I don't use ABA [Applied Behavioral Analysis] … I don't have training in ABA. I don't have specialized training in autism management. So, what I use is knowledge. I got from school and my experience working from one kid to another".* - **Psychologist 3**

*"Well, lack of skills, that was the first challenge because after my training you've learnt about psychopathologies, you've learnt about issues but when you go in the field and you have a child who is banging their head and you're working with them and you're like, okay what do I do here?*- **Psychologist 2**.

Additionally, HCPs in this study shared that lack of specialists in Uganda who are knowledgeable in ASD and SIBs management increased barriers in referral pathways. For example, the psychiatrist in this study shared that despite some parent's ability to pay for services, it is challenging to find appropriate specialists to provide services.

*"There are few specialists. Even if parents have money, they can't find therapists. One left the country, and no one replaced them…. it's always a struggle to get the specialists. So even if the parents had money, let's just assume, even getting specialists to give those services is quite tricky*. - **Psychiatrist**

Participant in this study revealed a shortage of resources needed to effectively manage SIBS among children diagnosed with ASD in Uganda. HCPs expressed frustration regarding unavailability of necessary tools and protocols needed to support their work with children presenting with SIBS.

*"…there are tools, [required tools to manage SIBs] I mean the resources. I don't have them; the resources are not available* -**Occupational Therapist**

Providers described feeling frustrated and overwhelmed when working with children who engage in self-injurious behaviors (SIBs), highlighting the emotional and physical toll of supporting these children. Many reported experiencing exhaustion and burnout due to the constant attention and vigilance required. One Speech and Language Therapist explained,

*"First of all, they are very hard to navigate. They are very hard to stop, and they're also very hard to work around with."*

*"By experience, it is very exhausting. It is very tiring because you have to be watching them. Most of their work [is] full time. You can only take a rest when they are actually sleeping."* -

These accounts suggest that limited staffing, high-intensity care demands, and the need for continuous monitoring place significant strain on providers, which may affect both their well-being and the quality of care they can deliver.

Providers also highlighted various challenges in working with parents of children with ASD engaging in SIBs including hope for quick fixes, agreement on treatment goals, inconsistencies in following suggested routines and limited understanding of ASD and SIBs. These challenges can hinder timely and effective interventions for children in this population.

*"The first thing is parents, but now I can understand they are desperate, and they want results there and then so they will scold you, they will blame you, they will do anything, they will be late, they will be inconsistent, but there's nothing to do."*- **Occupational Therapist 2**

*So, you would sit with them. Come up with a routine. But then they wouldn't follow it when you are not around. And sometimes you would have to train maids [stay-in nannies]. Because the maids stay at home with the kids. So, you are training the caregivers [stay-in nannies]. We would have sessions where we would train caregivers [stay-in nannies]. But then the caregiver [stay-in nanny] would say, the boss [parent] refused to do this.* **– Psychologist 2**

**Parent-related challenges:** Providers highlighted several challenges faced by parents of children with autism spectrum disorder (ASD) who present with self-injurious behaviors (SIBs), including financial constraints, stigma, exhaustion, and difficulties in understanding their child's behavior. Financial constraints often limited parents' ability to follow recommended care plans. A Pediatrician noted,

*"But sometimes the families are poor, they cannot afford all the things you're proposing and that can create a bit of frustration."* **Pediatrician**

*"A lot of people come into the clinic, and you tell them come after 2 weeks and they are like I don't have transport money. So sometimes maybe you want to see the kid after two weeks but the parent says I cannot afford and you know there is nothing you can do so you push it to a month."* **Psychiatrist**

Stigma also posed a significant challenge. One Psychiatrist recounted,

*"Highly stigmatized. One mother was battered because she had a child with autism. The father left, blaming her.*
*Psychiatrist*

*"We need to destigmatize autism—people still think it's a curse or bad parenting."* **Pediatrician**

These experiences underscore how social, economic, and cultural factors can exacerbate parental stress and hinder access to care, emphasizing the need for both community education and supportive interventions that consider families' realities.

In summary, the findings highlight several interrelated themes regarding healthcare providers' experiences with self-injurious behaviors (SIBs) in children with an ASD diagnosis in Uganda, including both their experiences in managing SIBs and the challenges they face while serving this population. The results underscore the complexity of caring for children with SIBs in the Ugandan context and reveal critical gaps in clinical practice, professional training, and systemic support. Addressing these gaps will require targeted policy interventions, increased resource allocation, and capacity-building initiatives to support providers and families, improve care quality, and reduce the burden of SIBs among children with ASD.

## Discussion

To our knowledge, no studies in Uganda have explored healthcare providers' experiences managing self-injurious behaviors (SIBs) among children with autism spectrum disorder (ASD). This study therefore fills an important gap by providing context-specific insights into how Ugandan providers understand and manage these complex behaviors. The findings reveal that managing SIBs is a complex and emotionally demanding process, impacted by providers' perceptions, therapeutic strategies, and the realities of practicing in resource-limited settings. Participants viewed SIBs as a means of communication and part of the repetitive and stereotypic behaviors characteristic of ASD.

Psychologists have long viewed SIBs as a response to a lack of control over intense negative emotions [18]. Children with SIBs often exhibit poor emotional regulation abilities and often lack outlets to express their distress; thus, self-injury becomes a process of expression and emotion management [19]. Consistent with this theoretical insight, HCPs who participated in the current study viewed SIBs as a means of communication, especially among children who are non-verbal. This perspective is consistent with the functional-behavioral approach that argues that SIBs serve a communicative and regulatory function [2,20]. Additionally, study participants described SIBs as resulting from sensory self-stimulation and boredom, a trend supported by previous studies that identified sensory dysregulation as a significant antecedent of SIBs [21]. Collectively, these results highlight the subtle understanding of the relationship between sensory, affective and communicative processes by HCPs working with children with ASD.

The management of SIBs in children with ASD relies on a multidisciplinary and evidence-based approach, which includes behavioral and, in some cases, pharmacological interventions [2,24]. There is also empirical evidence to support the assertion that structured behavioral techniques such as Functional Behavioral Assessment, reinforcement schedules, and social skills training produce the most significant changes in the frequency and severity of SIBs [24]. However, resource access, lack of sufficiently trained staff, and unavailability of special supplies are common challenges to the implementation of such protocols in low- and middle-income countries (LMICs), necessitating clinicians to use contextually adapted approaches to maintain safety and continuity of care [10,24]. Consistent with the previous scholars [10,24] clinicians in our study reported the use of combined pharmacological and behavioral interventions, along with innovative, locally adapted materials. These adaptations encompassed the reuse of locally available materials, such as using socks as gloves, applying masking tape to prevent self-injury, and making improvised splints, and modifying physical environments to minimize injury, thereby demonstrating practical ingenuity in resource-limited settings. The findings build on existing literature by illustrating how practitioners in Uganda utilize evidence-based interventions through improvisation and cultural adaptation when resources are limited. Additionally, most participants reported having little to no formal training and supervisory support when utilizing evidence-based practices (EBPs). This finding is consistent with the trends in LMICs at large, where awareness of EBPs is on the rise, but systematic training and fidelity checks are inadequate, [25]. Gaps in implementation fidelity may undermine the effectiveness of the therapeutic process and contribute to discouragement among providers and caregivers, particularly when clinical improvements are absent or inconsistent [26].

Management strategies of SIBs varied according to the nature and severity of the condition. Strategies such as physical restraint, the use of protective gear/aids, and modifying the child's immediate physical environment were prominent. Alongside these safety measures, providers implemented behavioral interventions targeting core areas of deficit in ASD. These included social skills training targeting areas such as communication and emotional regulation, which

are considered core to ASD and SIBs management [27–29]. For children with more severe SIBs and comorbid conditions such as Attention Deficit Hyperactivity Disorder (ADHD), medications such as risperidone were commonly used to manage irritability and aberrant behaviors prior to the introduction of therapy. These findings are consistent with previous studies where antipsychotics such as risperidone are reported as effective in managing irritability and aberrant behavior in children with autism [30,31]. However, their effectiveness in managing SIBs is inconclusive, with little research done to examine its effectiveness [32]. Additionally, the weak regulatory and medicine management system in Uganda contributes to unregulated prescriptions that can expose children to side effects and increase the risks of dependence on the drug [33].

HCPs reported financial constraints, societal stigma, and misconceptions about SIBs as significant challenges to their management of SIBs. Additionally, inconsistencies among parents and caregivers in adhering to recommended management strategies further contributed to provider frustration. Thus contributing to delayed treatment outcomes and persistent SIBs due to the limited availability of appropriate, specialized, and evidence-based interventions. Notably, even experienced practitioners reported considerable uncertainty when managing SIBs. The skills needed are frequently underemphasized in many training programs, leaving healthcare providers with gaps in skills, particularly in functional behavior assessment, functional communication training, and the implementation of evidence-based behavioral interventions. This contributes to reliance on short-term harm reduction strategies rather than long-term treatment goals. Despite the reported use of medication that aligns with international practice in complex cases, this raises concerns about informal prescription in an environment that lacks follow-up systems and integrated care pathways. In addition to professional gaps, participants highlighted social and economic conditions that undermine the effectiveness of the intervention strategies. Participants reported key factors that contribute to ineffective care, including parental and caregiver financial constraints, caregivers' lack of awareness, and the absence of organized referral systems. These structural constraints align with findings in the current literature on disability and mental health within low- and middle-income countries, where systemic constraints often limit the clients' efforts to achieve meaningful treatment outcomes. [34,35].

## Implications

These findings highlight critical implications for practice, training, and policy in low-resource settings. The shortage of trained healthcare providers (HCPs) and the absence of specialized care plans continue to hinder effective management of self-injurious behaviors (SIBs) [13]. The limited number of semi-skilled and inadequately equipped providers often resort to trial-and-error approaches using various behavioral modification techniques and psychopharmacological remedies [36,37]. Such practices may yield minimal benefit and, in some cases, pose risks to the well-being of children [38], while contributing to increased frustration and psychological distress among parents [39,40] and professional caregivers [41]. As a result, the mental health burden associated with poorly managed SIBs among children with Autism Spectrum Disorder (ASD), their families [42], and communities [43,44] may exceed the apparent prevalence of ASD itself.

To address these gaps, there is an urgent need to strengthen capacity for evidence-based behavioral interventions through contextually appropriate training and supervision systems. Developing practical, low-cost, and easily transferable intervention models adapted from successful examples in comparable low-resource settings [45,46] could significantly enhance the quality and sustainability of care provided by HCPs in Uganda and similar contexts.

## Limitations

This study has several limitations. First, all participants were based in urban areas; therefore, their experiences may not reflect those of healthcare providers (HCPs) working in peri-urban or rural contexts. Second, participants were drawn from facility-based settings such as hospitals, schools, and private practices and did not include perspectives from community-based providers or residential care facilities. Another limitation is information bias which may have occurred since the study relied on participant's self- reported experiences. Additionally, research bias was another possible

limitation as the researcher (HMM, HO and SO) have prior experiences working with children with SIBSs, thus the researchers prior experience may have influenced aspects of the research process, including data interpretation and theme development. Although these factors may limit the transferability of our findings, the study nonetheless provides valuable preliminary insights into the experiences of HCPs working with children on the autism spectrum who present with self-injurious behaviors (SIBs), offering a foundation for future research in this population. Furthermore, by including a diverse range of professionals such as occupational therapists, speech therapists, psychologists, pediatricians, and psychiatrists primarily from Uganda's capital city, we were able to capture a broader and more nuanced understanding of the key issues under consideration. Finally, as children with ASD and their parents were not included, the study cannot provide a complete representation of all experiences regarding SIBs, interventions, and related challenges.

## Conclusion

Our findings highlight the growing demand for skilled and well-equipped treatment teams to provide services for children diagnosed with Autism Spectrum Disorder (ASD) who present with self-injurious behaviors (SIBs) in Uganda. The provision of comprehensive, culturally sensitive interventions by well-trained healthcare providers is critical to addressing the diverse needs of both children and their caregivers. Furthermore, there is a need for additional research on the care and management of ASD, and SIBs in particular, in low-resource settings to deepen our understanding of the challenges faced by children, their families, and professional healthcare providers.

## Supporting information

**S1 Code. Qualitative codebook used for thematic analysis.**
(DOCX)

## Acknowledgments

We extend our gratitude to our co-creators of knowledge, the participant, for agreeing to take part in this study and sharing their experiences. We are also grateful to the following individuals who contributed to the successful completion of this manuscript: Sulaiman Kigozi, Stuart Ssebbaggala, Ms. Irene Nakalembe Kigozi, and the management of the Learner Support Unit at Kabojja International School for allowing us to use their facilities during the study.

## Author contributions

**Conceptualization:** Hillary Mugabo Mukula, Khamisi Musanje, Samuel Ouma.

**Data curation:** Hillary Mugabo Mukula, Harriet Opondo, Morris Ndeezi.

**Formal analysis:** Hillary Mugabo Mukula, Harriet Opondo, Morris Ndeezi, Khamisi Musanje, Samuel Ouma.

**Investigation:** Hillary Mugabo Mukula.

**Methodology:** Hillary Mugabo Mukula, Morris Ndeezi, Khamisi Musanje, Samuel Ouma.

**Project administration:** Hillary Mugabo Mukula.

**Resources:** Hillary Mugabo Mukula.

**Software:** Hillary Mugabo Mukula.

**Supervision:** Khamisi Musanje, Samuel Ouma.

**Validation:** Harriet Opondo, Morris Ndeezi, Samuel Ouma.

**Writing – original draft:** Hillary Mugabo Mukula, Morris Ndeezi, Anita Kateregga, Samuel Ouma.

**Writing – review & editing:** Hillary Mugabo Mukula, Harriet Opondo, Morris Ndeezi, Anita Kateregga, Samuel Ouma.

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
