## [Decision Letter · Decision Letter 0]

24 Feb 2026

PMEN-D-25-00531

“They are hard to navigate” exploring healthcare providers experiences managing Self-Injurious Behaviors among children with Autism Spectrum Disorder in Uganda

PLOS Mental Health

Dear Dr. Ndeezi,

Thank you for submitting your manuscript to PLOS Mental Health. We are sorry for the delay in reaching a decision. After careful consideration of the reviewer reports, we feel that the paper has merit but does not fully meet PLOS Mental Health’s publication criteria as it currently stands. Therefore, we invite you to submit a revised version of the manuscript that addresses the points raised during the review process.

Please ensure that you fully address all of the comments raised, which you can find at the end of this email and in the attached document.

We look forward to receiving your revised manuscript.

Kind regards,

Karli Montague-Cardoso

Staff Editor

PLOS Mental Health

Journal Requirements:

1. In the online submission form, you indicated that "Data is available upon request.".

3. Uploaded as supplementary information.

Additional Editor Comments (if provided):

Reviewers' comments:

Reviewer's Responses to Questions

**Comments to the Author**

1. Does this manuscript meet PLOS Mental Health’s publication criteria? Is the manuscript technically sound, and do the data support the conclusions? The manuscript must describe methodologically and ethically rigorous research with conclusions that are appropriately drawn based on the data presented.

Reviewer #1: Yes

Reviewer #2: Yes

2. Has the statistical analysis been performed appropriately and rigorously?

Reviewer #1: N/A

Reviewer #2: N/A

3. Have the authors made all data underlying the findings in their manuscript fully available (please refer to the Data Availability Statement at the start of the manuscript PDF file)?

Reviewer #1: Yes

Reviewer #2: No

4. Is the manuscript presented in an intelligible fashion and written in standard English?

Reviewer #1: Yes

Reviewer #2: Yes

5. Review Comments to the Author

Reviewer #1: Thank you for the opportunity to review this manuscript. The study addresses an important and timely topic, highlighting the urgent need for greater understanding management self-injurious behaviours (SIBs) among children with Autism Spectrum Disorder (ASD).

Data Collection

The author should elaborate on how data saturation was achieved in this study. A clear description of the processes and indicators used to determine saturation would strengthen the methodological rigour.

Ethical Considerations

The manuscript indicates that recruitment was conducted through the dissemination of study flyers within healthcare professionals’ (HCPs) professional associations. However, it does not specify whether formal permission was obtained from these associations to facilitate recruitment and data collection. It would be beneficial to provide the names of the associations involved and clarify the approval processes followed.

Furthermore, four interviews were conducted at participants’ workplaces. The manuscript does not explain how access to these workplaces was secured, nor whether institutional permission was granted. Clarification on the procedures followed to obtain organisational consent is necessary.

Table 2: Participant Demographics

In Table 2, the workplace for some participants is listed as “Clinical.” This designation is ambiguous and does not clearly convey the practice setting. It should be specified whether this refers to a private practice, hospital, clinic, or another healthcare environment.

Two participants reportedly had specialised training in ASD and the management of SIBs. However, the findings do not indicate whether the management strategies implemented by these trained participants differed in effectiveness from those without such training. The manuscript should clarify the impact of specialised training on the management of SIBs and whether this influenced outcomes or approaches to care.

Results and Thematic Analysis

Interpretation of the Fingings

A deeper level of thematic analysis is required to strengthen the study’s theoretical grounding and contribution to the body of literature. For example, the statement:

“Providers highlighted self-stimulating as being central to the development of SIBs. Notably, they identified boredom as contributing to persistent SIBs because most of these children have nothing to do,”

reflects surface-level experiential reporting from healthcare professionals and lacks sufficient theoretical engagement. The findings would benefit from more robust conceptual interpretation and linkage to existing theoretical frameworks and empirical scholarship.

Reviewer #2: Dear Authors

Abstract

• This qualitative study explored the experiences of healthcare providers managing self-injurious behaviors in children aged 5-12 years diagnosed with Autism Spectrum Disorder. Rather say This qualitative study explored the experiences of healthcare providers in managing self-injurious behaviours in children aged 5-12 years diagnosed with Autism Spectrum Disorder

• Kindly indicate the study design and how many healthcare providers were interviewed.

• What was the sampling strategy, like how were the healthcare providers selected?

• Three core themes emerged from the data, including perceptions of self-injurious behaviours, management strategies, and providers' experiences managing self-injurious behaviours. The study was on experiences, hence the need to have themes that only emerge from experiences, not perceptions. Kindly relook at the core themes.

• Keywords: Autism Spectrum Disorder; Self-Injurious Behaviours; healthcare Providers; Low-income Setting. The low income setting can not be a key word. Kindly add children, experiences and Uganda as key words. Also check how many keywords are required as per journal guidelines.

See other attached detailed comments

6. PLOS authors have the option to publish the peer review history of their article (what does this mean?). If published, this will include your full peer review and any attached files.

**Do you want your identity to be public for this peer review?** For information about this choice, including consent withdrawal, please see our Privacy Policy.

Reviewer #1: **Yes:** Gsakani Olivia Sumbane

Reviewer #2: **Yes:** Dr Geldine Chironda

Figure Resubmissions:

---

## [Decision Letter · Decision Letter 1]

22 Apr 2026

PMEN-D-25-00531R1

“They are hard to navigate” exploring healthcare providers experiences managing Self-Injurious Behaviors among children with Autism Spectrum Disorder in Uganda

PLOS Mental Health

Dear Dr. Ndeezi,

Thank you for submitting your revised manuscript to PLOS Mental Health. After careful consideration and reviewer feedback, we would like invite you to submit a revised version of the manuscript that addresses the final, but important, point raised by the reviewer (which you can find below).

We look forward to receiving your revised manuscript.

Kind regards,

Karli Montague-Cardoso

Staff Editor

PLOS Mental Health

**Journal Requirements:**

**Additional Editor Comments (if provided):**

Reviewers' comments:

Reviewer's Responses to Questions

**Comments to the Author**

1. If the authors have adequately addressed your comments raised in a previous round of review and you feel that this manuscript is now acceptable for publication, you may indicate that here to bypass the “Comments to the Author” section, enter your conflict of interest statement in the “Confidential to Editor” section, and submit your "Accept" recommendation.

Reviewer #2: (No Response)

2. Does this manuscript meet PLOS Mental Health’s publication criteria? Is the manuscript technically sound, and do the data support the conclusions? The manuscript must describe methodologically and ethically rigorous research with conclusions that are appropriately drawn based on the data presented.

Reviewer #2: Yes

3. Has the statistical analysis been performed appropriately and rigorously?

Reviewer #2: N/A

4. Have the authors made all data underlying the findings in their manuscript fully available (please refer to the Data Availability Statement at the start of the manuscript PDF file)?

Reviewer #2: Yes

5. Is the manuscript presented in an intelligible fashion and written in standard English?

Reviewer #2: Yes

6. Review Comments to the Author

**Reviewer #2:** Dear Authors.

Thank you for addressing most of the comments raised on the work submitted. However, the following comment was not addressed:

The Trustworthiness of

qualitative data in terms of

dependability, transferability

and confirmability was not

explained for this study. Only

credibility was described.

Kindly add.

7. PLOS authors have the option to publish the peer review history of their article (what does this mean?). If published, this will include your full peer review and any attached files.

**Do you want your identity to be public for this peer review?** For information about this choice, including consent withdrawal, please see our Privacy Policy.

Reviewer #2: **Yes:** Dr Geldine Chironda

**Figure Resubmissions:**

---

## [Editor Report · Decision Letter 2]

12 May 2026

“They are hard to navigate” exploring healthcare providers experiences managing Self-Injurious Behaviors among children with Autism Spectrum Disorder in Uganda

PMEN-D-25-00531R2

Dear Mr Ndeezi,

We are pleased to inform you that your manuscript '“They are hard to navigate” exploring healthcare providers experiences managing Self-Injurious Behaviors among children with Autism Spectrum Disorder in Uganda' has been provisionally accepted for publication in PLOS Mental Health.

Best regards,

Karli Montague-Cardoso

Staff Editor

PLOS Mental Health